# Transparent Reporting on Financial Assets as a Determinant of a Company's Value—A Stakeholder's Perspective during the SARS-CoV-2 Pandemic and beyond

**Beata Dratwińska-Kania** [1] , **Aleksandra Ferens** [1,*] **and Piotr Kania** [2]

1   Department of Accounting, Finance College, University of Economics in Katowice, ul. 1 Maja 50, 40-287 Katowice, Poland
2   Department of Public Finance, Finance College, University of Economics in Katowice, ul. 1 Maja 50, 40-287 Katowice, Poland
*   Correspondence: aleksandra.ferens@ue.katowice.pl

**Abstract:** Background: Socio-economic changes prompt companies to disclose their sustainable development activities in the reporting, showing that they balance three capitals—economic, environmental, and social. On the other hand, while formulating strategies and goals, they consider the company's widely understood environment, where its stakeholders are essential. As a result, the transparency and usefulness of the reported information are limited. Methods: The study employed financial statements' content analysis and a statistical method (rank Spearman correlation, Shapiro–Wilk test). The percentage of change in critical areas for reporting transparency on financial assets was analyzed, before and during the SARS-CoV-2 pandemic. Results: The research indicated that the identified critical reporting areas concerning financial assets showed a greater value change during the pandemic. Correlations between the accounting value of the company and the same critical reporting areas were significant. Conclusion: It has been shown that larger companies can use more accounting policy instruments; therefore, the reporting transparency on financial assets is potentially lower for these companies. The transparency of the surveyed corporate reports during the pandemic was lower.

**Keywords:** financial assets; transparency; sustainable development; company's value; income smoothing; SARS-CoV-2 pandemic

## 1. Introduction

In recent years, the balanced and sustainable development of companies has become the dominant trend in formulating corporate development strategies, as well as strategies for publishing the content of financial and non-financial statements. The concepts of sustainable development presented in the literature are not consistent. Each of them may be the subject of various controversies and, at the same time, inspiration for an exciting discussion. Discrepancies in interpreting this term affect not only the differences in the methods of implementation but also the possibilities and degree of achieving the expected results. Each approach adopted has far-reaching practical consequences and therefore raises various doubts as it is assessed concerning other interests. According to Piontek, previous research provides 43 definitions of sustainable development [1] (p. 18), while Dobson has identified over 300 definitions and interpretations of the sustainable development concept [2] (p. 76). Amid the advent of the United Nations' Sustainable Development Goals (SDGs), global sustainability discourse has progressed to a point where it is inseparable from the firm's role [3]. Balanced and sustainable development of a company is a development in which human capital is treated as the greatest good. The permanent improvement of the quality of life of present and future generations through economic activity (the functioning of physical, financial, natural, and intellectual capital) should consider the preservation

of environmental quality. It should allow for short-term and long-term optimization of human and natural capital utilization [4] (p. 31). Therefore, a definition can be adopted, according to which the concept of balanced and sustainable development serves to steer economic (business), ecological, and social development, in which the integrating criterion is the improvement of the quality of life of current and future generations and thus the improvement of the lives of those with whom it interacts [5].

Many scientists consider this concept to be the only rational idea for human survival on earth [6]. Another proposed definition of balanced and sustainable development of companies delineates it as the social and economic development of companies, enabling today's pursuit of aspirations and making profits, without violating the possibility of pursuing aspirations and making profits in the future [7].

According to one of the most critical theories in the field of sustainable development, the company should balance three capitals: economic, environmental, and social [8,9]; thus, when formulating strategy and goals, the company must take into account not only profit but also other capitals, including the widely understood environment of the company (the environment where the company operates). One of the elements of the company environment is the recipient of reporting information, also known as the stakeholder. The idea that the environment could be considered a stakeholder was first put forward by Freeman [10]. Furthermore, Starik, for example, has said that the current anthropocentric definition of the concept "stakeholder" may be expanded to "any naturally occurring entity which affects or is affected by organizational performance" (quoted from Gibson [11] (p. 16)).

When the stakeholder is well-informed, we achieve balance in part of one of the capitals—the broadly understood environment as the company's environment, contributing to the implementation of sustainable development. Therefore, it can be said that economic information is an important element in building sustainable development, and a lack of reliable and credible information hampers this development. In recent years, public stakeholders and disclosure standards have gained considerable power in their ability to drive trends toward more sustainable business practices [12].

The information recipient is not a homogeneous category. It includes managers of the entity, members of supervisory boards, owners, contractors, and other potential users of reporting information. Examples of various stakeholder concepts are presented in Table 1.

**Table 1.** Definitions of the stakeholder concept.

| Source | Stakeholder's Definition |
|---|---|
| Freeman, Reed [10]; Freeman [13] | Identified groups or entities that may impact the achievement of the organization's goals or that are impacted by the goals achieved by the organization. |
| Klefsjö, Bergqu, Garvare [14] | Those actors who provide the necessary resources or support the organization and whose resources can be withdrawn if their needs or expectations are not met. |
| EFQM [15] | All those interested in the organization, its activities, and its achievements. |
| NIST [16] | All groups that are or may be affected by the activities or success of the organization. |

Source: own elaboration.

Sometimes, there is a conflict of interest between stakeholders and policy for creating financial statement tools used in shaping the content of financial and non-financial statements. Research showed, for example, that companies in Japan, listed companies in Iran and Teheran, and US banks use policy for creating financial statement tools to smooth out income [17]. Other studies indicated that management employs policy for

creating financial statement tools to influence stock prices, particularly the current share price [18]. Executives may be involved, for example, in income smoothing when they are concerned about maintaining their positions or avoiding reducing their expected financial benefits [19–22].

Income smoothing most often occurs when it is probable that the company will report a loss or when it is at risk of achieving profit threshold values. The literature also indicates that the recognition of additional compensation (e.g., share options for management) is also a symptom of income smoothing [23]. Share prices are believed to be higher when income is smoothed out and unpredictable gains lead to lower stock prices [18]. In other words, the capital market appreciates companies that report stable profits more because such profits are easier to forecast. The reason for income smoothing may also be access to external financing and its conditions. Smoothing income can affect access to capital and its cost.

Therefore, the potential recipient of the reports is a potential addressee of the company's policy for creating financial statements, which disrupts the sustainability of environmental capital and the implementation of sustainable development. The applied instruments of policy for creating financial statements (in particular, earnings management, smoothing income) may make it challenging to receive reporting information and to understand the regularities occurring in the field of a company's investment decisions. Such actions can be considered destructive to value [24,25].

It is also believed that corporate governance weakens the impact of income smoothing on the choice of accounting principles; in particular, it is believed to weaken the impact of the bonus plan on the choice of accounting principles. The more corporate governance factors are implemented in the company, the more it can be expected that the accounting principles selected by the management (accounting policy regarding the company and its environment) are not motivated by the management's interest but are in the interest of the company [26].

Therefore, there is a need to identify critical areas in corporate reporting which may indicate the instruments used in the policy for creating financial statements. Such effort was undertaken by the authors, analyzing information on financial assets from the financial statements (Section 2.4.). The authors consider the dissemination of research in this area helpful in reducing the damaging practice of companies in the excessive use of instruments of the policy for creating financial statements in the future, as well as developing stakeholders' awareness of these instruments used by companies, thus contributing to the implementation of sustainable development.

The conducted research indicated the tools used in the policy for creating financial statements recognized in the reporting of large Polish companies in the years 2018–2021 and showed their relationship with the company value, calculated with accounting measures. The authors did not find in the literature comprehensively identified critical reporting areas that could prove the company's policy for creating financial statements in the field of financial assets. The authors, therefore, consider this a theoretical achievement. In addition, the literature does not agree on whether reporting transparency goes hand in hand with the company value. Hence, the authors put forward their research in the international discussion in this area.

The theoretical goals of the work are:

1. Identification of critical reporting areas on financial assets for the implementation of the policy for creating financial statements. These areas are to prove reporting transparency.
2. Indicating a voice in the discussion on the relationship between the transparency of reporting information on financial assets and the company value calculated using accounting methods, such as the value of total assets, fixed assets, current assets, equity, and other comprehensive income.

The empirical goals of the work are:

1. Identification and measurement of changes in the value of critical reporting areas on financial assets;

2. 　　Measurement of correlations between critical reporting areas for financial assets;
3. 　　Measurement of the correlation between the critical areas of financial asset reporting and the company value, calculated with accounting measures.

## 2. Theoretical Framework and Literature Review

### 2.1. Reporting Transparency as One of the Generators of a Sustainable Company Value

The sustainable value management concept is a hybrid one. It was created to express the coincidence of the value management concept with the corporate social responsibility and sustainable development concepts [27]. The sustainable value management concept is founded on a combination of pro-value ideas, sustainable development, and management ethics [28]. In this concept, achieving social goals is closely related to reaching economic goals that foster economic and social justice [29]. For example, keeping stakeholders properly informed regarding the company's activities, including investment operations related to financial assets, can be considered a social goal. We may note that, along with the development of pro-value ideas, the information approach, including the provision of information to stakeholders, has also evolved and gained importance. According to behavioral theory, giving a priority role to stakeholders contributes to taking such actions by the company that may affect its positive perception, which, in turn, will increase its competitive position and translate into financial success. The idea of sustainable development implies that it should ensure that society and future generations meet ecological, economic, and socio-cultural standards.

In light of the company's focus on sustainable value management, there is a need to change the scope and quality of reporting by companies, primarily public ones, taking into account standards and guidelines. Reporting transparency is becoming increasingly important, thanks to which the stakeholder can be better informed about the company's activities, which, as argued in Section 1, translates into sustainable development. The greater the reporting transparency, the more difficult it is to employ earnings management.

Transparency, or clarity, lucidity, or translucency, are terms initially used to describe a characteristic feature of an object. This term has also been assigned to activities, organizations, systems, and initiatives that support transparency, i.e., to the appropriate disclosure of achievements, tasks performed, and compliance with the standards of behavior in the economy and society. The best-known initiative of this type is Transparency International, an independent and non-governmental organization that exposes and combats corrupt practices, especially in the public sector. The concept of transparency is applied to various objectives, such as, for example, transparency of the management board's activities, transparency of the application of control procedures, or decision-making transparency. The authors propose to define the transparency of financial statements, particularly the transparency of information on financial assets.

The concept of transparency is evolving. Transparency reveals three aspects: transparency as a public value adopted by society to counter corruption, transparency synonymous with open decision making by governments and non-profit organizations, and transparency as a complex tool of good governance in programs, policies, organizations, and nations [30]. Stiglitz drew attention to the need for transparency in the information disseminated and openness in government. Citizens have a fundamental right to open and transparent information about the activities of the government as public trustees [31]. To make a distinction, de Haan and Amtenbrink [32] defined transparency as the degree of actual understanding by external representatives of the policy and decision-making process (since the research was conducted on the central bank, this concerned monetary policy). Transparency can also be defined as a lack of information asymmetry: if there were transparency, the private sector and the central bank would have virtually the same information. There is also a fairly large body of research on central bank transparency, including defining the transparency concept concerning the central bank's information policy [33]. Winkler [34] proposed a more comprehensive definition of some aspects of central bank transparency and introduces the concept of the optimal amount of information,

i.e., the information level at which the benefit resulting from it balances the costs associated with its acquisition, analysis, and interpretation. He defined this concept as information effectiveness. Winkler [34] enriched the definition of transparency by including three aspects of the information transfer process. In addition to openness in providing information, this author considered clarity, informative effectiveness, and honesty. The considerations presented testify to the importance of the discussed research problem and its inclusion in the streamlined dialogue in published research.

The evolution of transparency is also reflected in its definition. According to Barth and Schipper [35], comprehensibility as a feature of financial statements can be considered the equivalent of transparency. Transparency, understood as a clear, open, and understandable presentation of information on the transactions and performance of an economic entity, can be found in the standards of the Global Reporting Initiative—GRI [36]. Tamowicz and Dzierżanowski (cited by [37]), listing the attributes of transparency, indicated the availability of information and its completeness, relevance, quality, and credibility. Transparent information, made available through reporting to a wide range of stakeholders, should therefore enable its unbiased interpretation. Thus, there is no doubt that this feature is difficult to apply in practice. The reporting transparency addressed to many recipients is complicated to determine (and to assess). The multitude of information recipient groups and their different information needs make it extremely difficult to define a set of reporting information that everyone would consider transparent. Transparent information should be determined by adapting it to the needs of the environment. This results in the need to constantly monitor the information needs of stakeholders and adjust the scope and manner of information presented to them.

In considering reporting on the risk of financial assets as a company value generator, according to the authors, the understanding of transparency by Bushman et al. [38] (p.207) takes on special relevance. They defined it as the availability of specific information on the activities of a business entity to people outside. The authors accept this definition as fundamental and binding for further research. In what follows, the authors also assume transparency in disclosing information to individual groups of stakeholders as comprehensiveness. The main purpose is to reduce the asymmetry of information between management, owners, and other users of financial statements. This is supposed to result in potentially better use of available resources by stakeholders.

Transparency is a culture of coexistence between the company and its environment. It often results in disclosing critical information to critical competitors, but on the other hand, transparency determines market trust. Determining what should be the quantitative and qualitative scope of information presented in corporate reporting to be considered as ensuring public trust is not an easy task, but it is vital because access to this information is recognized around the world as the basis for creating value [39].

Numerous studies have been conducted on the transparency of financial statements. Among them, the following should be mentioned:

- Research conducted by Lapointe-Antunes et al. in Switzerland in 1997–2001 [40] showed that increasing disclosures in the income statement, while applying IFRS (International Financial Reporting Standards) or Generally Accepted Accounting Principles (GAAP), increases the transparency of the financial situation and, as a result, reduces the scope of shaping financial results.
- Kang and Hoong [41] studied the relationship between the scope of disclosures assessed, based on the economic development of the parent country of an international company and the compliance of accounting information. The research showed that the relationship between the US stock price and home country accounting information varies with local economic development and is related to the transparency of disclosures. The greater the transparency of disclosures in companies operating in developed economies, the greater the consistency of accounting information compared to companies from emerging economies.

- Research by Pawnall and Schipper [42] proved that there are different degrees of disclosure transparency; similar events and transactions are differently recognized and reported.
- In their studies, Lundholm and Myers [43] showed no clear evidence that the value of current profits increases as the quality of disclosures increases. According to them, actions disclosed by companies give reliable and consistent information not reflected in current profits but in current share prices.
- Verrecchia [44] demonstrated that the market response to disclosure is an increasing function of its quality (i.e., the accuracy of the information disclosed). More transparent (and therefore greater-quality) disclosures allow investors to better interpret data (such as profits).
- Rajan and Zingales [45] researched the information asymmetry between the company's management and external investors. They indicated that this asymmetry is by and large greater in countries with a less developed economy, and it also affects the development of external capital markets.
- Dratwińska-Kania [46] proved the existence of interdependence between the transparency level of the investment fund, measured with an original model, and the variables describing the behavior of stakeholders, i.e., the change in the number of participation units in the financial year and the following year as well as the change in the value of participation units in the investment fund.
- Szewieczek, Dratwińska-Kania, and Ferens analyzed the relevance of integrated reporting as a tool of transparency and accountability in state-owned companies in a sample of European state-owned companies in 2013–2017. They revealed that companies preparing an integrated report present a highly detailed level of business model reporting elements [47]. On the other hand, the business model is a significant step in the holistic presentation of the company's value, reducing the information gap experienced by stakeholders and increasing the transparency of market disclosures [48].

The authors in this study intend to examine the reporting transparency on financial assets in the form of identification and analysis of reporting areas that are crucial for the implementation of the policy for creating financial statements.

### 2.2. The Relationship between Reporting Transparency on Financial Assets and a Company Value

The category constituting the basis for evaluating the way of managing the company and the primary determinant of decision making in all key areas of activity is value. Value is the conformity between the existing situation and what should be [49]. Zarzecki [50] defined value as a feature of an object by which a given object is perceived as desirable, respected, useful, or important. Thus, values are standards of what is desirable. There are many definitions of value in the economic literature, but what does not change is the assumption that companies function to create value. Value is a set of tangible and intangible benefits that meet the requirements of stakeholders in a timely, effective, and efficient manner [51]. According to Karmańska [52], value in the business context means money if it is given to an object engaged in business activity, then the value is measurable in financial terms, and the object meets the value when it meets the expectations regarding economic benefits. It is also possible to identify the company's value and its creation as the strategic quintessence of the company's bundle of objectives, generated for all interested entities from the company's tangible and intangible resources [53] (p. 8). On the other hand, Karmańska [52] recognized that the economic value and its measurement is an area where the third general accounting paradigm should be considered—besides the socio-economic paradigm in accounting and the balance sheet valuation paradigm. Many studies confirm that reporting transparency is of great importance in creating the company's value, particularly the tools used in the policy for creating financial statements, such as income smoothing.

For example, Susanto and Pradipta [54] proved that the impact of company value on income smoothing is significant and positive. The situation is different depending on



the size of the company: here, its impact on income smoothing is also significant, but negative. The study argued that companies that create value for investors will try to retain them and apply methods of income smoothing, which is supposed to convince them of a stable increase in profits and retain investors. On the other hand, large companies are less likely to engage in income smoothing, as they are convinced that investors will continue to engage anyway. Similarly, the DLT Swastika study [55] showed that there is a significant and negative relationship between the size of the company and earnings management. Prasetya and Rahardio [56] also demonstrated in their research that some of the companies listed on the Indonesian Stock Exchange used income-smoothing practices. The company's size as the analyzed factor did not affect income smoothing. Research by Desiyanti and Desaputra [57] was conducted on the Indonesian market, and it was found that the size of the company and net profit do not significantly affect income smoothing. Similar research among companies in India was carried out by Bora [58] and proved that income smoothing does not depend on the size of the company. Habib [23] studied companies operating in Bangladesh and found that 46 companies out of a sample of 107 engaged in at least one type of income-smoothing behavior. The article proved, among other things, that smaller companies were more involved in income smoothing. The impact of income smoothing on the value of the company in the regulated security market was examined in the article by Abogun et al. [59]. The study indicated that most companies in the Nigerian market engaged in income smoothing, and this practice had a significant negative impact on the company value.

On the other hand, research by Wijay and Mauren [60] indicated that the size of the company has a significant, positive impact on income smoothing. Similarly, Ernayani et al. [61] analyzed the factors affecting the practice of income smoothing. The research found that the size of the company significantly affects income smoothing. Statistical analysis by Anwar and Chandra [62] on a sample of 29 companies listed on the Indonesian Stock Exchange only partially showed that the size of the company significantly impacts income smoothing. The study's results by Indrawan et al. [63] are also positive for this relationship. The authors proved that company size has a direct positive impact on income-smoothing practices. The 2013–2015 Indonesian market of companies listed on the stock exchange was characterized by the fact that the larger the company was, the more often income-smoothing practices were used. Lambert [64] believed that the incentive to smooth income is, among others, company size. Graham et al. [18] showed that managers manage earnings to influence the current share price (so there is a positive relationship with company size).

The presented body of literature indicates that the issues raised are important and often taken up in research. There is no specific position on the impact of the transparency of disclosures on the value/size of the company. Therefore, the study examines the correlation between the critical reporting items of financial assets, in the case of which the instruments of the policy for creating financial statements could be used (including earnings management), i.e., elements contributing to the transparency of information on financial assets and the company value, calculated with accounting measures. The following research hypothesis was put forward:

**Hypothesis 1 (H1).** *It is assumed that the reporting transparency on financial assets is correlated with the company value, calculated with accounting measures.*

The study will examine whether larger companies use more critical reporting areas on financial assets, potentially limiting financial statements' transparency.

### 2.3. The Relevance of the SARS-CoV-2 Pandemic Period for the Information Disclosure on Financial Assets

The impact of the SARS-CoV-2 pandemic on financial statements and earnings management is a current and vital research issue. The International Organization of Securities Commissions [65] recognized that the SARS-CoV-2 pandemic was significant for information disclosure by companies. Hariadi and Kristanto [66] argued that the SARS-CoV-2

pandemic had an impact on earnings management among companies in Indonesia. Similarly, Arnold [67], Laux and Leuz [68], and Ozili [69] believed that during crises, including pandemics, companies could use accounting techniques to improve a deteriorating balance sheet and statement of comprehensive income. Lassoued and Khanchel [70], on a sample of 2,031 companies listed in 15 European countries, examined the tendency to manage income during the pandemic and in the previous period. The research results showed the reduced credibility of financial statements during the SARS-CoV-2 pandemic. Nurcahyono, Sukesti, and Alwiyah [71] studied whether the SARS-CoV-2 pandemic affected the quality of financial statements and what factors affected the quality of government financial statements. It was found that the pandemic affected this quality, with environmental uncertainty being the primary negative factor. Bose, Shams, Ali, and Mihret [72], in a sample of 4278 companies from 47 countries, examined the relationship between the impact of the SARS-CoV-2 pandemic on the change in the company value and whether the results are affected by the implementation of the sustainable development principles. It was found that companies operating in countries with a greater impact of SARS-CoV-2 experienced a greater decrease in company value. On the other hand, among companies implementing the principles of sustainable development, the negative impact of SARS-CoV-2 on the company value was not clear. It was argued that the activities of companies in the field of sustainable development mitigate the negative impact of SARS-CoV-2 on the change in the company value. Gilchrist [73] thought similarly—companies operating within a sustainable development framework outperform others. Qiu et al. [74] surveyed listed companies in China's hospitality industry and found that during the pandemic, investors are responding positively to CSR activities that help protect communities, employees, and customers from the virus.

However, some researchers demonstrated opposite trends during the pandemic—companies reported a decrease in profit, which was justified by the pandemic situation [75,76], or the pandemic period was considered less favorable for earnings management due to a more thorough examination of financial statements by auditors [77,78].

Based on the above studies, it can be concluded that there are no clear results as to the impact of the SARS-CoV-2 pandemic on the financial statements. Therefore, the authors of the study put forward the following hypothesis:

[H2] It is assumed that during the SARS-CoV-2 pandemic, critical reporting areas for financial assets most exposed to earnings management practices, i.e., potentially reducing the transparency of financial statements, show a greater percentage of change than the change in these critical areas in the times before the SARS-CoV-2 pandemic.

The authors of the study aim to analyze the percentage of change in critical reporting areas, i.e., the change in the classification group of financial assets and the valuation model, the recognition of financial assets valued using the own model, other comprehensive income on financial instruments, and the net value of reversed impairment cost and income. These areas are believed to have the greatest impact on the transparency of financial asset reporting. This issue is discussed in Section 2.4.

### 2.4. Possibilities of Adopting the Policy for Creating Financial Statements concerning Financial Assets

The policy for creating financial statements toward the environment constitutes a set of instruments for a differentiated reporting approach to events and conditions in an economic entity. What distinguishes it from accounting policy regarding the company is the desire to influence the stakeholders of the financial statements, thanks to the applied rights of choice and freedom of action, which are allowed under the applicable accounting regulations. The concept of the policy for creating financial statements is understood as a set of lawful decisions of an economic entity, aimed at shaping the following in the financial statements: financial result, assets, equity, liabilities, and other items to ensure optimal implementation of economic assumptions [79]. These are all activities undertaken during the financial year and during the preparation of the annual financial statements, aimed at influencing the

recipients' assessment of the balance sheet and motivating them to adopt the behavior desired by the entity [80]. It should be noted that the accounting legal regulations and English-language literature use the term accounting policy (not the policy for creating financial statements), which is equated with accounting principles. The accounting policy is defined, among other things, as the method of accounting adopted by the entity, within the framework set by law, consisting of the selection and consistent application by the entity of specific accounting principles, rules, methods, and procedures which, in the opinion of the management board, are appropriate to the company's situation and best represent its financial picture [81]. There is also the concept of earnings management, including income smoothing, which, according to the authors, should be included in the instrumentation of the policy for creating financial statements.

When analyzing the definitions quoted, the common feature of these definitions is that they refer to the entity's choice of alternative accounting solutions for the financial statements permitted by the regulations. They pay particular attention to the quality of information presented in the financial statements and the impact on the recipients. Appropriate conduct of the policy for creating financial statements creates the possibility of creating the image of the company, which leads to the achievement of its goals and economic assumptions and consequently the possibility of creating the company value.

The following instruments of the policy for creating financial statements, regarding the recognition and valuation of financial assets in the account books, can be indicated as follows [46,82]:

- estimating asset impairment;
- estimating future cash flows using the fair value method when there is no active market for the valued asset;
- estimating the interest rate, discounting future cash flows in the valuation, using the fair value method when there is no active market for the valued asset—this is a market-dependent variable—the real rate of return on theoretically risk-free investments, the expected inflation rate over the duration of the investment, or the premium for investment risk incurred;
- estimating the streams of return in the valuation using the adjusted purchase price method, e.g., based on financial results, future cash flows, dividend payments, interest payments, or increase in initial capital;
- selecting the fair value estimation model when there is no active market for the financial asset held;
- selecting the period of historical observations, based on which forecasts are made in the adopted model when there is no active market for the financial asset held;
- making approximations in the estimated valuation of financial assets, especially in the case of high-value financial assets;
- applying subjective judgments, e.g., in terms of estimating future cash flows—the most probable, average, or another justified variant;
- changing the classification group of financial assets, especially when they are valued as other comprehensive income.

Based on the conducted theoretical studies, it was found that it is sometimes difficult to state the actual information value of financial statements with certainty, since a large part of financial assets is valued based on prospective assumptions and estimates. In such circumstances, it is much easier to use the instruments of the policy for creating financial statements and much more difficult to notice what the person preparing financial statements wanted to hide. In addition, changes in the assumptions made in the valuation may affect the comprehensive income statement, and therefore it can also be used for earnings management (income smoothing). Recognition of changes in the value of financial assets as other comprehensive income distracts the stakeholder because it does not affect financial costs/income. Based on the above, four critical reporting areas were selected for further analysis:

- change in the classification group of financial assets and the valuation model;

- recognition of financial assets valued using their own model;
- other comprehensive income on financial instruments;
- the net value of reversed impairment cost and income.

These items are examined for changes during and before the pandemic, and correlations between these four critical reporting areas are checked to see if their utilization practices are mutually reinforcing. Then, the correlations that occur between these critical reporting areas, showing reduced transparency of the financial statements and the company value, calculated by accounting measures, are examined.

## 3. Materials and Methods

The study was conducted in three parts. First, the percentage change in critical areas for the transparency of financial asset reporting, indicated in Section 2.4, was analyzed during the SARS-CoV-2 pandemic and before the pandemic to prove the second research hypothesis and attain the first empirical goal. The results of the analysis are presented in Tables 2–5. Second, it was examined whether there are correlations between these critical reporting areas, which could indicate that these potential tools of the policy for creating financial statements have been used complementarily. This would indicate potentially lower transparency of the financial statements. Third, the correlations between the critical areas of financial assets reporting and the company value, calculated with accounting measures, such as the value of total assets, the value of fixed assets, the value of current assets, the value of equity, and the value of other comprehensive income, were examined. The study aims to prove the first hypothesis and to attain the second and third empirical goals.

**Table 2.** Change values in the classification group and valuation model of financial assets.

| Company | Classification Group and Valuation Model Changein PLN Million | | | | |
| --- | --- | --- | --- | --- | --- |
| | 2018 | 2019 | Change 2019/2018 | 2020 | 2021 |
| Lotos s.a. | 2667.40 | - | Na | - | - |
| PKN Orlen s.a. | 11,516.00 | - | Na | - | - |
| PGE s.a. | 33,142.00 | - | Na | - | - |
| Tauron s.a. | 28,604.62 | - | Na | - | - |
| Enea s.a. | 19,144.31 | - | Na | - | - |
| JSW s.a. | 2179.700 | - | Na | - | - |
| KGHM s.a. | 9527.00 | - | Na | - | - |
| Orange s.a. | 909.00 | - | Na | - | - |
| Cyfrowy Polsat s.a. | 12,126.10 | - | Na | - | - |
| Kogeneracja s.a. | 169.12 | - | Na | - | - |
| CCC s.a. | 579.40 | - | Na | - | - |
| Asseco s.a. | 2083.40 | - | Na | - | - |
| Grupa Azoty s.a. | 4188.56 | - | Na | - | - |
| PKP Cargo s.a. | 1094.9 | - | Na | - | - |
| Rafako s.a. | 74.90 | - | Na | - | - |
| Wawel s.a. | 72.41 | - | Na | - | - |
| Agora s.a. | 610.94 | - | Na | - | - |
| Asseco see s.a. | 599.57 | - | Na | - | - |
| Grupa Kęty s.a. | 373.20 | - | Na | - | - |
| Kruk s.a. | 3049.86 | - | Na | - | - |

Source: own elaboration.

**Table 3.** Values and changes in the value of financial assets measured using their own model.

| Company | Recognition of Financial Assets—Own Model in PLN Million | | | | | | |
| --- | --- | --- | --- | --- | --- | --- | --- |
| | 2018 | 2019 | Change 2019/2018 | 2020 | Change 2020/2019 | 2021 | Change 2021/2020 |
| Lotos s.a. | 16.20 | 52.70 | 225% | 558.40 | 960% | 1019.50 | 83% |
| PKN Orlen s.a. | 1480.00 | 1950.00 | 32% | 2853.00 | 46% | 5331.00 | 87% |
| PGE s.a. | 346.00 | 551.00 | 59% | 1376.00 | 150% | 2612.00 | 90% |
| Tauron s.a. | 4.18 | 19.46 | 366% | 72.76 | 274% | 1.26 | −98% |
| Enea s.a. | 7274.08 | 7421.35 | 2% | 7633.06 | 3% | 7236.01 | −5% |
| JSW s.a. | 1833.300 | 1934.50 | 6% | 498.90 | −74% | 519.20 | 4% |
| KGHM s.a. | 863.00 | 275.00 | −68% | 498.50 | 81% | 518.40 | 4% |
| Orange s.a. | - | 0.40 | na | - | na | - | na |
| Cyfrowy Polsat s.a. | 0.60 | 0.40 | −33% | - | na | 13.40 | na |
| Kogeneracja s.a. | 0.94 | 4.44 | 372% | - | na | - | na |
| CCC s.a. | - | - | na | 810.10 | na | 810.10 | 0% |
| Asseco s.a. | 27.50 | 12.00 | −56% | 11.30 | −6% | 11.30 | 0% |
| Grupa Azoty s.a. | 67.66 | 7.68 | −89% | 49.97 | 551% | 61.43 | 23% |
| PKP Cargo s.a. | 8.00 | 12.20 | 53% | 4.90 | −60% | 4.90 | 0% |
| Rafako s.a. | - | - | na | - | na | - | na |
| Wawel s.a. | 133.42 | 151.73 | 14% | 142.77 | −6% | 199.33 | 40% |
| Agora s.a. | - | - | na | - | na | - | na |
| Asseco see s.a. | 1.58 | 0.12 | −92% | 0.07 | −42% | - | na |
| Grupa Kęty s.a. | 5.89 | 1.55 | −74% | 2.34 | 51% | 1.87 | −20% |
| Kruk s.a. | 38.80 | 34.03 | −12% | 29.62 | −13% | 13.80 | −53% |

Source: own elaboration.

**Table 4.** Values and changes in other comprehensive income on financial instruments.

| Company | Other Comprehensive Income on Financial Instruments in PLN Million | | | | | | |
| --- | --- | --- | --- | --- | --- | --- | --- |
| | 2018 | 2019 | Change 2019/2018 | 2020 | Change 2020/2019 | 2021 | Change 2021/2020 |
| Lotos s.a. | - | - | na | - | na | −12.30 | na |
| PKN Orlen s.a. | −171.00 | −141.00 | 18% | −383.00 | −172% | −173.00 | 55% |
| PGE s.a. | −138.00 | −86.00 | 38% | −267.00 | −210% | 659.00 | 347% |
| Tauron s.a. | −24.30 | 15.18 | 162% | −103.17 | −780% | 0.46 | 100% |
| Enea s.a. | −68.83 | −1.69 | 98% | −108.86 | −6341% | 265.20 | 344% |
| JSW s.a. | 18.00 | 38.90 | 116% | 27.20 | −30% | −23.30 | −186% |
| KGHM s.a. | 155.00 | −391.00 | −352% | −692.00 | −77% | −280.00 | 60% |
| Orange s.a. | −13.00 | −27.00 | −108% | −13.00 | 52% | 376.00 | 2992% |
| Cyfrowy Polsat s.a. | - | - | na | - | na | - | na |
| Kogeneracja s.a. | - | - | na | - | na | - | na |
| CCC s.a. | - | - | na | - | na | - | na |
| Asseco s.a. | - | - | na | - | na | 0.70 | na |
| Grupa Azoty s.a. | - | - | na | - | na | 3.94 | na |
| PKP Cargo s.a. | −22.50 | 10.10 | 145% | −50.50 | −600% | 9.00 | 118% |
| Rafako s.a. | - | - | na | - | na | - | na |
| Wawel s.a. | - | - | na | - | na | - | na |
| Agora s.a. | - | - | na | - | na | - | na |
| Asseco see s.a. | - | - | na | - | na | - | na |
| Grupa Kęty s.a. | −7.32 | 5.52 | 175% | 0.66 | −88% | −0.69 | −205% |
| Kruk s.a. | −11.13 | 3.95 | 135% | −14.86 | −476% | 32.67 | 320% |

Source: own elaboration.

**Table 5.** Value and changes in the net value of reversed impairment cost and income for financial assets.

| Company | Net Value of Reversed Impairment Cost and Income in PLN Million | | | | | | |
| | 2018 | 2019 | Change 2019/2018 | 2020 | Change 2020/2019 | 2021 | Change 2021/2020 |
|---|---|---|---|---|---|---|---|
| Lotos s.a. | - | −30.80 | na | −9.40 | 69% | 0.80 | 109% |
| PKN Orlen s.a. | 1113.00 | 1332.00 | 20% | 225.00 | −83% | 1390.00 | 518% |
| PGE s.a. | - | - | na | - | na | - | na |
| Tauron s.a. | - | - | na | - | na | - | na |
| Enea s.a. | −207.47 | −359.39 | −73% | −3757.26 | −945% | - | na |
| JSW s.a. | - | - | na | −12.70 | na | −4.40 | 65% |
| KGHM s.a. | 1450.00 | 156.00 | −89% | 21.00 | −87% | 807.00 | 3743% |
| Orange s.a. | - | - | na | - | na | - | na |
| Cyfrowy Polsat s.a. | - | - | na | - | na | - | na |
| Kogeneracja s.a. | - | - | na | - | na | - | na |
| CCC s.a. | - | - | na | −252.60 | na | −24.70 | 90% |
| Asseco s.a. | 8.00 | 0.90 | −89% | - | na | 7.30 | na |
| Grupa Azoty s.a. | - | −0.06 | na | −5.50 | −9067% | - | na |
| PKP Cargo s.a. | −22.50 | 10.10 | 145% | −50.50 | −600% | 9.00 | 118% |
| Rafako s.a. | 0.10 | - | na | 0.55 | na | - | na |
| Wawel s.a. | 1.27 | 0.03 | −98% | −0.74 | −2567% | −0.07 | 91% |
| Agora s.a. | 0.10 | 0.10 | 0% | 0.05 | −50% | 0.05 | 0% |
| Asseco see s.a. | −1.27 | −0.16 | 87% | 9.77 | 6206% | 6.82 | −30% |
| Grupa Kęty s.a. | - | - | na | - | na | - | na |
| Kruk s.a. | - | - | na | - | na | - | na |

Source: own elaboration.

The study employed content analysis of financial statements and a statistical method (rank Spearman correlation, Shapiro–Wilk test). The percentage of change in critical areas for reporting transparency on financial assets was analyzed, before and during the SARS-CoV-2 pandemic.

## 4. Results

The empirical study covered the separate financial statements of large Polish companies listed on the Warsaw Stock Exchange which are not financial institutions, apply the accounting principles contained in IFRS, and have financial assets in their portfolios. A total of 20 companies were selected for the study, and it was considered that such a sample is representative of Polish conditions, because other companies reviewed did not report any information relevant to the study. Four years were selected for the survey: 2018 and 2019, when there was no SARS-CoV-2 pandemic, and 2020 and 2021 with the SARS-CoV-2 pandemic.

Table 2 shows the change in value in the classification group of financial assets and the change in the valuation model. Table 3 shows an analysis of the percentage change in one of the critical areas: the recognition in the financial portfolio of financial assets valued using their own model in 2018–2021, i.e., those for which there are no market prices, and their value is individually estimated. Table 4 presents the values and value changes in other comprehensive income related to financial instruments. Table 5 shows the values and changes in the net value of reversed impairment cost and income for financial assets. Since not all companies reported on the critical areas mentioned, a "-" sign was used when there was a lack of reporting information. In the absence of information, it is also impossible to calculate the change in the value of these critical areas, in which case the "na" sign was placed, i.e., not applicable.

In the case of the classification group and valuation model change, the reporting information concerns only one year, 2018, when the new IFRS changed the classification of financial assets. In other years, companies did not report such information; thus, it is

impossible to analyze whether there were more reclassifications of financial assets during the pandemic years or not. On the other hand, the lack of reported information may indicate limited reporting transparency.

In the case of this area, it should be noted that already in the first year of the pandemic, most companies showed significantly increased values of financial assets valued using their own model. In the second year of the pandemic, this trend continued. A change in the value of financial assets, valued with their own model, can be easily obtained by changing the valuation parameters, i.e., interest rate, expected cash flows, and others, as indicated in Section 2.4. The change in value in this critical area was more significant in the years of the pandemic than in the years before. This may indicate the tools of the policy for creating financial statements used in connection with the valuation of these securities, because, in most cases, it was an increase in the value of financial assets, and this proves that during the pandemic, companies could generate potentially less transparent financial statements to hide failures in implementation plans due to the pandemic. This research part proves the second hypothesis and implements the first empirical goal.

In the case of the reporting area of other comprehensive income of financial instruments, the change in the value of this area in 2020 was large, in most cases higher than in previous years, but negative. This may prove that companies hedged themselves against the uncertain times of the pandemic by not reducing impairment losses for financial assets, but rather other comprehensive income, which is an item that does not affect stakeholders as much as financial costs. In the subsequent year 2021, the value of these write-downs also changed significantly, more than the change in 2019/2018, but here a large part of the change in value was positive. This may indicate that previous decreases in value were used to recognize increases in the value of assets in the next period and thus improve financial results. It can therefore be concluded that, in the case of this area, there is a high potential for companies to use the tools of the policy for creating financial statements more during the years of the pandemic than in the years before. This research part proves the second hypothesis and implements the first empirical goal.

In the case of changes in the net value of reversed impairment cost and income for financial assets, a large proportion of companies did not include in their financial statements information necessary for research. This information is sporadic; there are no data at all in a given company and no value in selected years. As a result, it was not always possible to calculate changes in the value of this area. Only in the five companies marked in the table was it possible to calculate the sought changes in value. This sample is too small to draw general conclusions. In these five cases, the changes during the pandemic were usually more significant than in the pre-pandemic period, which confirms the current trend that, during the pandemic, companies potentially used more tools of greater value in the policy for creating financial statements, which made the reporting transparency during the pandemic potentially lower than before. In addition, it can also be stated that the lack of reporting in this area is also information for the stakeholder, putting the transparency of the financial statements in an unfavorable light. This research part proves the second hypothesis and implements the first empirical goal.

The Shapiro–Wilk test was carried out to examine the distribution of correlation variables in the second and third parts of the research. Since the variables do not have a normal distribution, the rank Spearman method of correlation was chosen to study the relationship between the variables, both for computing the interdependence between the critical areas of financial assets reporting and for the relationship between these critical areas and the company value, calculated with accounting measures. The significance level of the study was set at 0.1. The sample size is 20, the critical value is 0.3804. Statistically significant correlations are marked in the tables. The results of the correlation calculations for all model components in individual years are presented in Tables 6 and 7. The study is intended to prove the first research hypothesis and to achieve the second and third empirical goals.

**Table 6.** Correlations between critical reporting areas on financial assets.

| Item | Recognition of Financial Assets—Own Model | | | | Other Comprehensive Income on Financial Instruments | | | | Net Value of Reversed Impairment Cost and Income | | | |
|---|---|---|---|---|---|---|---|---|---|---|---|---|
| | 2018 | 2019 | 2020 | 2021 | 2018 | 2019 | 2020 | 2021 | 2018 | 2019 | 2020 | 2021 |
| classification group and valuation model change | 0.5087 | - | - | - | −0.4089 | - | - | - | −0.1300 | - | - | - |
| recognition of financial assets—own model | | | | | −0.1415 | −0.1839 | −0.4047 | −0.1084 | 0.0669 | 0.0885 | −0.3593 | −0.0672 |
| other comprehensive income on financial instruments | | | | | | | | | 0.3166 | −0.1185 | −0.2270 | −0.1789 |

Source: own elaboration.

**Table 7.** Correlations between critical areas of financial assets reporting and company value, calculated using accounting measures.

| Item | Classification Group and Valuation Model Change | | | | Recognition of Financial Assets—Own Model | | | | Other Comprehensive Income on Financial Instruments | | | | Net Value of Reversed Impairment Cost and Income | | | |
|---|---|---|---|---|---|---|---|---|---|---|---|---|---|---|---|---|
| | 2018 | 2019 | 2020 | 2021 | 2018 | 2019 | 2020 | 2021 | 2018 | 2019 | 2020 | 2021 | 2018 | 2019 | 2020 | 2021 |
| Total Assets | 0.8497 | - | - | - | 0.4764 | 0.6047 | 0.5420 | 0.5811 | −0.4160 | −0.4681 | −0.6597 | 0.0836 | 0.0382 | 0.0075 | −0.0342 | 0.2243 |
| Fixed Assets | 0.8801 | - | - | - | 0.5042 | 0.6133 | 0.5195 | 0.5564 | −0.3982 | −0.4100 | −0.6645 | 0.0955 | −0.0147 | 0.1080 | −0.0072 | 0.2477 |
| Current Assets | 0.6367 | - | - | - | 0.4741 | 0.6443 | 0.6305 | 0.6632 | −0.2445 | −0.4238 | −0.5650 | −0.0621 | 0.1195 | −0.0150 | −0.0859 | 0.0918 |
| Kapitał Własny Equity | 0.8131 | - | - | - | 0.4616 | 0.6457 | 0.4979 | 0.5559 | −0.3446 | −0.4420 | −0.6399 | 0.0604 | 0.0628 | 0.1215 | 0.0076 | 0.2625 |
| Other TOTAL Comprehensive Income | 0.1758 | - | - | - | 0.5421 | 0.4708 | −0.2603 | 0.5663 | 0.2370 | −0.2085 | 0.0042 | −0.1588 | 0.1621 | 0.0898 | 0.3185 | 0.1607 |

Source: own elaboration.

In the analysis of Table 6, it should be stated that some of the cross-correlations between the critical reporting areas on financial assets are statistically significant, and some are not. This proves that reporting transparency is potentially at risk. Still, it would be worse if there were more significant correlations (in this case, it could be said that these areas are used as complementary instruments of the policy for creating financial statements, which would lower reporting transparency about financial assets). There is a significant correlation between certain areas, i.e., (1) the classification group and valuation model change and (2) the recognition of financial assets valued using their own model (when there are no market prices) in 2018. Perhaps some financial assets have been reclassified to a group valued using their own model, but such detailed information is unfortunately unavailable in the reports. There is also a significant correlation in 2018 between other areas, i.e., (1) the classification group and valuation model change and (3) other comprehensive income on financial instruments, and this is a negative correlation. This may indicate potential transfers of financial assets, measured in other comprehensive income to the group measured in profit or loss, i.e., for example, the release of positive revaluations of financial assets retained in other comprehensive income and their recognition as financial income. Negative correlations also occur in the year of the 2020 pandemic, namely between (2) the recognition of financial assets measured by their own model (when there are no market prices) and (3) other comprehensive income on financial instruments, and also between (2) the recognition of financial assets measured by their own model and (4) the net value of reversed impairment cost and income. In the first case, there may be a potential threat that the more financial assets are valued using their own model, the lower the other comprehensive income on financial instruments, i.e., the risk of reducing other comprehensive income for assets valued using their own model. Perhaps companies use the valuation with their own model to write down the value of financial assets to establish a "provision" for future, uncertain times (pandemic period). Another significant correlation can be interpreted in a similar way, which proves a decrease in the net value of reversed impairment cost and income with an increase in assets valued using their own model. The conclusions of Table 6 are reinforced by the fact that, based on Table 3, the value of assets valued using their own model increased significantly during the pandemic years.

Table 7 presents the correlations between the critical reporting areas on financial assets and the company value, calculated with accounting measures. The table contains many significant correlations for critical areas, i.e., (1) the classification group and valuation model change, (2) the recognition of financial assets valued using their own model, and (3) other comprehensive income on financial instruments (in the third case, the correlations are negative). In the case of the first critical area, correlations were calculated only for 2018, when the new IFRS on financial instruments entered into force, changing the existing classification of financial assets. Therefore, companies had to report changes. In the following years, companies did not report on the reclassification of financial assets or a change in the valuation model, which is the basic tool of the policy for creating financial statements. In the critical area (4) the net value of reversed impairment cost and income, no significant correlations with any accounting measure of company value were noted. It can therefore be concluded that in most cases, for the first three critical reporting areas, significant correlations with the company value, calculated with accounting measures, were noted. This confirms the first hypothesis and the implementation of the second theoretical goal and the third empirical goal. The study proves that larger companies can use more policy instruments for creating financial statements.

## 5. Discussion

Proponents of the stakeholder theory claim that building and maintaining lasting relationships with stakeholders has a positive effect on the process of building the company's value; therefore, it is beneficial for the company to consider the interests of a wide range of stakeholders. Such proponents included Freeman [10], Clarkson [83], Wheeler and Sillanpaa [84], Carroll and Buchholtz [85], Adamczyk [86], Ząbkowicz and Pietras [87].

There are also opponents of the stakeholder theory who believe that managers' primary goal is to build company value for owners and shareholders. Proponents of this concept place owners as the most favored interest group because the use of company resources for the benefit of all interest groups undermines property rights, denying owners the ability to make decisions about their property [88–90].

The authors take the position that, unfortunately, the opponents of the stakeholder theory win. In the research, companies with a higher book value used a higher value of items critical for the policy for creating financial statements, so they could use more (more significant number, higher amount) of these tools in the scope of providing information on financial assets. The presented research confirms the theory of opponents of maintaining relations with stakeholders. Companies were guided by their own benefit in the form of the possibility of using profit management, rather than the interest of stakeholders. This is consistent with most of the literature studies included in Sections 2.1 and 2.2, although there is a discussion on this issue in the literature. In addition, profit management practices may have been used to a greater extent during the SARS-CoV-2 pandemic, as during this period the value and change in the value of sensitive reporting areas of financial assets were more significant than before the pandemic. This is also in line with most of the literature studies presented in Section 2.3, albeit there is also some discussion on this point.

Another question discussed in the literature and practice also concerns income smoothing, namely whether this reporting practice benefits the company and its stakeholders. On the one hand, the market expects stable profits and rewards them, which is a positive phenomenon [24,91,92]; however, some actions of managers, although within the limits of the law, may distort the true and fair view of the company and be considered as damaging the value [24,25,93]. The line between these two arguments is vague, which is why the authors of the study take an ideological position—income should not be "smoothed", as this may distort the true and fair view principle.

There is also a discussion in the published research about the amount of information disclosed and its usefulness. Francis [94] showed in his research that the inclusion of detailed profit and loss statements increases the absolute market response to profit statements. It is also suggested that investors find more detailed and transparent disclosures helpful in making decisions. It represents the position that more information increases its usefulness. The opposite category to increasing disclosures in financial statements is their brevity. Slimming down financial statements has been a significant accounting narrative in recent years. Therefore, when choosing the appropriate level of transparency of financial statements, one should consider the usefulness of disclosures and the clarity of the financial statements, related to limiting the amount of information presented. Transparent financial statements are rich in additional disclosures and concise, so transparency is a compromise between these two categories. The information provided on financial assets is economical; therefore, the practice of increasing its amount should bring benefits to stakeholders. Similarly, transparency is a search for a compromise between the broadly understood comparability of the financial statements and taking into account the specificity of the company and highlighting information relevant to this specificity, at the expense of information irrelevant to this specificity. The authors of this study take the position that in the case of financial assets under Polish conditions, a larger amount of reporting information will increase its usefulness. The conducted research showed that companies do not disclose information on, for example, changes in the classification group of financial assets and changes in the principles of balance sheet valuation of these components. There is no information on which financial costs arose in connection with the reclassification of financial assets. The information on financial assets is fragmented; thus, the practice of extending it should bring benefits to stakeholders.

It can therefore be concluded that reporting transparency benefits both stakeholders and the company. Literature research on banks in China indicates that maintaining reporting transparency goes hand in hand with the financial stability of banks [95] as well as with the reduction in low-quality assets [96].

## 6. Conclusions

The main issue considered was the reporting transparency on financial assets for the stakeholder. This was deemed an important issue from the sustainable development standpoint. Furthermore, four critical areas were identified that could become a tool of the policy for creating financial statements, which could also reduce the transparency of financial statements.

The study analyzes the change in critical reporting areas, i.e., the classification group of financial assets and valuation model change, the recognition of financial assets valued using their own model, other comprehensive income on financial instruments, and the net value of reversed impairment cost and income. It was found that during the pandemic, these areas showed a greater change in value, which may indicate reduced transparency of the financial statements due to the use of the policy for creating financial statements to a greater extent during the pandemic than before the pandemic. In addition, the correlations between these critical reporting areas were examined, and it was found that there were not many statistically significant relationships. This is good news for stakeholders; this means that only in some cases could these critical areas be used complementarily, which could potentially intensify the effects of the applied policy for creating financial statements on the transparency of financial statements. The correlations between the book value of the company and these critical reporting areas were most significant for three areas: change of financial asset classification group and valuation model, the recognition of financial assets valued using their own model, and other comprehensive income on financial instruments. Therefore, it can be concluded that larger companies may use more tools of the policy for creating financial statements because, for the first three critical reporting areas for financial assets, significant correlations with the size of the company were noted. Thus, for larger companies, the transparency of information on financial assets is potentially lower, as companies can use the policy for creating financial statement tools in these areas.

**Author Contributions:** B.D.-K., A.F. and P.K. conceived of the presented idea; P.K., B.D.-K. and A.F. developed the methodology and framework; B.D.-K., P.K. and A.F. performed the study and drafted the manuscript; A.F., B.D.-K. and P.K. designed the figure; B.D.-K., A.F. and P.K. encouraged and supervised the study and analysis. All authors discussed and contributed to the final manuscript. All authors have read and agreed to the published version of the manuscript.

**Funding:** This research was funded within a subsidy for the maintenance and development of the research potential of the University of Economics in Katowice, Poland.

**Institutional Review Board Statement:** Not applicable.

**Informed Consent Statement:** Not applicable.

**Data Availability Statement:** The data that support the findings of this study are available from beatakania@ue.katowice.pl upon reasonable request.

**Acknowledgments:** The authors acknowledge three anonymous reviewers whose remarks and comments significantly contributed to the development of the presented article.

**Conflicts of Interest:** The authors declare no conflict of interest.

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
