# Peer review of "Transparent Reporting on Financial Assets as a Determinant of a Company’s Value—A Stakeholder’s Perspective during the SARS-CoV-2 Pandemic and beyond"

_sustainability, doi:10.3390/su15032065_

Round 1
Reviewer 1 Report
The file attachment is below

Reviewer 2 Report
Dear author, I really appreciate your efforts, however, I believe that while building the arguments for transparent reporting on financial assets you have missed some important studies which you must include. The main question that you have addressed and the main theoretical contribution needs further elaboration. Kindly refer to the available literature for example,
ashir, U., Khan, S., Jones, A. et al. Do banking system transparency and market structure affect financial stability of Chinese banks?. Econ Change Restruct 54, 1–41 (2021). https://doi.org/10.1007/s10644-020-09272-x
The topic is original and need of the time, but I was unable to find the relevance with regard to theory and practice in the objectives of the study. Please clarify what exactly specific gap your study will fill in the field? There is relatively less critical analysis and less emphasis to the subject area compared with other published materials.
In the methodology specifically you need improvements in the form of linking your methodology with the latest studies that have been conducted in the field. Kindly follow and link your methodology with the suggested literature;
Bashir, U., Yu, Y., Hussain, M., Wang, X., & Ali, A. (2017). Do banking system transparency and competition affect nonperforming loans in the Chinese banking sector?. Applied Economics Letters, 24(21), 1519-1525.
To improve the article especially with reference to theoretical and practical significance, I am suggesting you another citation regarding innovation that can significantly improve your work in terms of contribution to the body of knowledge. Please read and cite the following articles to improve the study;
Bashir, U. (2022). Shadow banking, political connections and financial stability of Chinese banks: an empirical investigation. Applied Economics Letters, 1-5.
I strongly recommend you to use these article to strengthen your study especially in the introduction and literature section as far as its contribution to the body of knowledge is concerned. Secondly link your methodology and identify how you improve with the available literature. Like the findings for innovation is hardly linked with prior literature in the article and also likewise for the sustainability, I strongly recommend you to link your findings with the suggested findings to improve the quality of your paper. Then it will then have a significant impact in the body of knowledge.
Reviewer 3 Report
The subject treated is original and innovative and the authors indicate their contribution to actual knowledge.
Previous works on the same topic are well mentioned.
The demonstrations are carried out in a logical way.
The abstract summarizes well the main points treated in the article.
Tables and diagrams are essential for a better understanding and are all called in the text.
The conclusions are correct and justified by the demonstration.
The article has the potential to be published, but the authors have to operate some revisions in order to give more consistency to the manuscript.
Despite the paper's aim being an interesting one, some points need to be considered for publishing it. The paper should be briefer in the first part or organized in a more consistent and clear way. It could be reduced or better represented in the framework part, where there are explained the concept of transparency and value. Moreover, the reference to the Covid pandemic represents an important part. For this reason, it could be mentioned in the title to better highlights the purpose of the study and its potential applications.
Round 2
Reviewer 2 Report
most of the comments are addressed.